# Role of Knowledge, Sociodemographic, and Behavioral Factors on Lifetime HIV Testing among Adult Population in Nepal: Evidence from a Cross-Sectional National Survey

**DOI:** 10.3390/ijerph16183311

**Published:** 2019-09-09

**Authors:** Bimala Sharma, Eun Woo Nam

**Affiliations:** 1Department of Community Medicine, Gandaki Medical College, Pokhara 33700, Nepal; 2Yonsei Global Health Center, Yonsei University, Wonju, Gangwon-Do 26493, Korea; 3Department of Health Administration, Graduate School, Yonsei University, Wonju, Gangwon-Do 26493, Korea

**Keywords:** HIV testing, sexual partner, mobility, HIV knowledge, demographic health survey, Nepal

## Abstract

Human immunodeficiency virus (HIV) testing is important to HIV prevention, treatment, and care. We aimed to assess the role of sociodemographic, behavioral factors and HIV knowledge on HIV testing among people aged 15–49 years in Nepal. The 2011 Nepal Demographic and Health Survey data was used for secondary data analysis. Herein, 9843 women and 3017 men who had experienced coitus were included. The respondents were asked if they underwent HIV testing and received the test results in their lifetime. Chi-square test and multivariate logistic regression analysis were applied at 5% level of significance. Adjusted odds ratios and 95% confidence intervals were computed separately for women and men. Of the total, 18.0% of men and 7.4% of women had been tested for HIV in their lifetime. As compared to the age of 15 to 24 years, males aged 25 to 29 years were more likely to report, whereas females aged 35 to 49 years were less likely to report HIV testing. Lower caste groups had more likelihood of reporting HIV testing than the other caste in both sexes. The odds of being tested for HIV were significantly higher among those who had higher education in both sexes. There was significant positive association between HIV testing and economic status in males whereas this association was reverse among females. The male respondents who spent more than one month away from home in the last 12 months were 1.68 times more likely to have been tested for HIV in their lifetime. Having multiple sexual partners was associated with higher odds of testing for HIV in both sexes. Having comprehensive HIV knowledge was independently associated with the reporting of higher odds of HIV testing in females. Promotion of HIV testing should consider sociodemographic factors, sexual behavior, and imparting comprehensive HIV knowledge.

## 1. Introduction

Human immunodeficiency virus (HIV) in Nepal was first reported in 1988. The estimated HIV prevalence among people age 15 years and above is below 1%, however there is concentrated epidemic among high risk groups. Heterosexual transmission is a major mode of transmission. The estimated prevalence is highest in the age group of 25–49 years. Male to female ratio of annual number of new infections is 2.02 to 1, and the average number of new infections per day is 3 in Nepal. For achieving universal access to prevention, treatment, and care, HIV Testing and Counseling has been a strategic focus in the national response to HIV since 1995. Now, HIV testing is provided free of charge through government and nongovernment HIV testing sites all over Nepal [1]. Nepal is a land locked country of South Asia that boarders with China in the North, and with India in the East, the West and the South with area of 147,181.00 sq. km. Nepal has a total population of 28 million, with an average annual population growth rate of 1.35%. It is a developing country with a human development index of 0.548, and a life expectancy of 69.6 years [2,3].

Voluntary HIV screening is cost-effective even in health care settings in which HIV prevalence is low. Several modifiable and non-modifiable factors were consistently identified as being associated with HIV testing [4]. Some of these factors include sociodemographic factors [5,6,7,8], mobility [9], and the type and number of sexual partners [4,5,10].

HIV infection via seasonal labor migrants from India is the key factor contributing to the HIV epidemic in Nepal [11]. High-risk sexual behaviors are also prevalent among the general population in Nepal [10,12,13]. Nepal has recently launched the new National HIV Strategic Plan (2016–2021) with the 90-90-90 goal of UNAIDS, that is, by 2020, 90% of all people living with HIV will know their HIV status; 90% of all people with diagnosed HIV infection will receive sustained antiretroviral therapy; 90% of all people receiving antiretroviral therapy will have viral suppression [14].

A theoretical framework considered for the study is, in addition to individual behavioral change and knowledge, the broad determinants of sexual behavior, such as gender, poverty, and mobility, should be focused in public health interventions [15]. Despite a growing body of research on female sex workers (FSWs), injecting drug users (IDUs), male who have sex with male (MSMs), transgender, and their role in HIV epidemic [16,17], HIV-related knowledge and the behaviors of sexually and socially active general population in Nepal have remained poorly investigated. Given that studies conducted among the general population are rare, the general population’s knowledge and practice on sexual behavior, especially on HIV testing is largely unexplored [18]. Thus, the study aimed to identify the role of sociodemographic and behavioral factors, and HIV knowledge associated with HIV testing among males and females aged 15–49 years in Nepal.

## 2. Materials and Methods

### 2.1. Study Area, Design, and Sampling

This study used the secondary data from the 2011 Nepal Demographic and Health Survey (NDHS). NDHS 2011 is a nationally representative cross-sectional survey conducted under the government of Nepal. It is a part of worldwide Demographic and Health Survey (DHS) Program of USAID, which is carried out at every five years. A publicly available data set from the DHS MEASURE website was obtained [19]. By merging relevant information from the data sets of women’s and men’s questionnaire, a new data set was created. The questionnaire was developed by DHS program and used internationally; for the survey, the questionnaire was translated into three major language of Nepal and was pretested. The details of the questionnaire and procedures can be found in the publicly available website and survey report of the 2011 NDHS [19,20]. The NDHS used a multistage cluster sampling procedure for data collection. For the study, 9843 women and 3017 men who reportedly had experienced sexual intercourse were identified for the analysis.

### 2.2. Study Variables

HIV testing was measured and categorized as (i) “yes = 1” if respondents ever experienced HIV test and received the test result, and (ii) “no = 0” if otherwise [8].

### 2.3. Measurement of HIV Knowledge

Ten questions relevant to HIV knowledge were selected from the DHS questionnaire. Correct responses were coded as “1,” whereas incorrect or uncertain responses were coded as “0.” Items were summed to obtain the HIV knowledge score, with higher scores indicating more knowledge about HIV transmission and prevention; and it was used as continuous variables [21] (Table 1).

### 2.4. Explanatory Variables

For the measurement and classification, most of the variables were categorized as they were categorized in the 2011 NDHS data set [20] (Table 2). The economic status of each household was calculated by the 2011 NDHS using the principal component analysis of more than 40 asset variables [22]. The calculated wealth quintiles were provided in the data set which were used in the analysis. Mobility was measured by asking as “In the last 12 months have you been away from home community for more than one month at a time?” and was categorized into ‘yes’ and ‘no’. Ethnicity was measured as upper caste, lower caste and other caste as it was done in the previous article that used NDHS, 2011 [23].

### 2.5. Data Analysis

The data from the 2011 NDHS were analyzed using the Statistical Package for Social Sciences (SPSS) version 20.0. Chi-square test and independent sample t test were utilized to assess the association between the explanatory variables and HIV testing. Multivariate logistic regression analysis was also conducted. Separate analysis was performed for males and females. Adjusted odds ratio and 95% confidence interval are computed for each explanatory variable. The level of significance was set at 5% for all the analyses. Multicollinearity was checked; variance inflation factor (VIF) of each variable with HIV knowledge was less than 2.

### 2.6. Ethical Consideration

The 2011 NDHS was approved by Nepal Health Research Council and Ethical Review Board of ICF Macro International. The data set is completely anonymous and distributed in the public domain of DHS without any identifiable information about the participants. These anonymous data have no restrictions on use.

## 3. Results

### 3.1. General Characteristics of the Study Population

Of the total respondents, 76.5% were females and 23.5% were males. Among males, 22.6% were in the age group of 15 to 24 years, 87.0% were currently married, and 69.0% lived in rural areas. In addition, 18.5% of the males belonged to the poorest and 26.1% belonged to the richest wealth quintile. Moreover, 15.7% of the male respondents had no formal schooling. Of the males, 31.9% were involved in agriculture, and 4% were unemployed. Furthermore, 39.8% of the male respondents were from the upper caste, and 84.6% were Hindus. In the last 12 months, 22.6% of the male respondents had been away from home for more than 1 month. Of the total, 39.3% of the male respondents had multiple sexual partners in their lifetime. The mean HIV knowledge score was 7.8 out of a maximum of 10. Age at first sex was 19.5 years. Of total males, 18.0% had been tested for HIV in their lifetime (Table 2).

Among females, 25.1% were in the age group of 15 to 24 years, 96.0% were currently married, and 72.5% were from rural areas. Of the females, 20.4% belonged to the poorest and 22.7% belonged to the richest wealth quintile. Moreover, 47.7% of the female respondents had no formal schooling. Of the females, 59.3% were involved in agriculture, and 20.8% were unemployed. Furthermore, 40.7% of the respondents were from the upper caste, and 85.8% were Hindus. In the last 12 months, 10.7% of the female respondents had been away from home for more than 1 month. Of the total, only 4.5% of the female respondents had multiple sexual partners in their lifetime. The mean HIV knowledge score was 7.5 out of a maximum of 10. Age at first sex was 17.4 years. Of total females, 7.4% had been tested for HIV in their lifetime (Table 2).

### 3.2. HIV Testing by Sociodemographic and Behavioral Characteristics

Table 3 shows the proportions of HIV testing by sociodemographic and behavioral characteristics, and the association of these variables with HIV testing among males and females. All variables, except religion and marital status, were significantly associated with HIV testing among males. Similarly, all variables, except marital status, were significantly associated with HIV testing among females.

Table 4 shows multivariate logistic regression analysis of factors associated with HIV testing. Among males, participants aged 25 to 29 years were more likely to report HIV testing than the respondents of age group 15 to 24 years (AOR, 1.46; CI, 1.03–2.06). Lower caste groups were 1.53 times more likely to report HIV testing in their lifetime than the other caste in males. Furthermore, the odds of being tested for HIV was significantly higher among those who had secondary (AOR, 2.82; CI, 1.67–4.78) and higher education (AOR, 2.85; CI 1.58–5.15) as compared to those who had no schooling in males. As compared to the males who belonged to the poorest group, male respondents who were from middle (AOR, 1.99; CI, 1.28–3.07), richer (AOR, 1.68; CI, 1.06–2.66) and the richest (AOR, 2.35; CI, 1.44–3.83) group reported higher odds of HIV testing. The male participants who spent more than one month away from home in the last 12 months were 1.68 times more likely to have been tested for HIV in their lifetime. The males who reported more than one sexual partner in their lives were 1.33 times more likely to have been tested for HIV in their lifetime. However, comprehensive HIV knowledge was not independently associated with the reporting of HIV testing in males.

Respondents aged 35 to 39 and 40 to 49 years were 47% and 53%, respectively less likely to report HIV testing among females. As compared to the female of other caste, female belonging to upper caste and lower caste were 1.94 and 3.57 times, respectively more likely to report HIV testing. Females with higher education reported higher odds of HIV testing as compared to the females with no formal education (AOR, 1.59; CI, 1.06–2.37). Women who were involved in a formal job were more likely to have been tested for HIV, as compared to unemployed women (AOR, 1.58; CI, 1.20–2.07). However, women belonging to poorer, middle, richer groups were 49%, 57%, and 33% less likely to report HIV testing as compared to the females belonging to the poorest group. Increased age at first sexual intercourse was associated with increased probability of having been tested for HIV in their lifetime among women. Meanwhile, among women, more comprehensive HIV knowledge was independently associated with the reporting of higher odds of HIV testing.

The Hosmer and Lemeshow Test showed that all variables entered into model were fit with the models for males and females (*p* > 0.05).

## 4. Discussion

The lifetime prevalence of HIV testing was 18% among males and 7.4% among females. At present, HIV testing is provided free of charge through government and nongovernment HIV testing sites all over Nepal. However, this free service is underutilized, indicating as one of the important challenges [14]. In Nepal, the HIV testing program is generally being promoted to high-risk groups, such as SWs, CSWs, IDUs, and migrant workers. Additionally, HIV screening in pregnant women has also been practiced since 2005, and the prevention of mother to child transmission (PMTCT) services have been gradually expanding to more health facilities [13]. The HIV testing is underutilized because either people do not perceive themselves at being at risk of HIV, they do not know about the test, or they are afraid of the consequences of testing positive [24]. A study among returned migrants from gulf countries reported that only 7% had ever heard about VCT services in Nepal [25]. A study using the 2011 NDHS shows that a high-risk situation for HIV infection was prevalent among 3.6% of adult males in Nepal [10]. Despite the risk of transmission, people do not perceive themselves being at risk of infection, and most of them were unaware of the VCT service.

In the current study, males were more likely to report being tested for HIV than females. Even though the HIV screening of pregnant women has been started, the PMTCT program is still not universally applied in all health facilities in Nepal [14]. Thus, the HIV testing among females is still significantly lower than that among males. In contrast, women were more likely to be tested for HIV than men in most of the previous studies [1,5,8]. The most important indicator for HIV testing among women is the antenatal care (ANC) [8]. Thus, the HIV testing in ANC package can be one of the useful strategies to cover a large population for HIV screening in Nepal.

The study shows that males aged 25 to 29 years were more likely to report HIV testing. It also showed that the risk of HIV infection was significantly associated with a younger age in Nepal [15]. Similarly, females belonging aged 35 to 39 years and 40 to 49 years were less likely to report HIV testing, as compared to the respondents of 15 to 24 years in the study. In contrast, studies conducted in high HIV-prevalent African countries show that the odds ratio of HIV testing increased, as the age of the respondents increased [6].

No significant difference in HIV testing was noted among the participants living in either the rural or urban area in the study. It shows that availability of the testing services has not influenced the testing behavior of people to the great extent.

In Nepal, the caste system is one of the important factors influencing health behaviors and service utilization. Castes were divided into three categories: upper caste, lower caste, and others [23]. In the current study, lower caste groups were more likely to be tested for HIV than the other caste group among both males and females. This study recommends that all programs related to HIV testing promotion should consider the upper caste and other castes with increased emphasis.

In the current study, having a higher level of education was significantly associated with an increased likelihood of being tested for HIV in both sexes. Similarly, a higher education level was significantly associated with higher odds of HIV testing, according to a study by Takarinda et al. (2016). Education is the most important cross-cutting issue associated with the increased service utilization and safe behaviors, including sexual behavior. It is considered as an important social determinant of health status of a community and a country [26]. Thus, investing more in education might benefit for a higher utilization of HIV testing, thereby preventing the increased HIV prevalence in Nepal.

In the study, occupation was not associated with HIV testing among males. However, service holder females were more likely to be tested for HIV as compared to unemployed. Likewise, unemployed participants had 63% lower OR of having used HIV testing services among MSMs in Nepal [16]. Occupation is generally related to income levels as well exposure to information and social networking, possibly influencing the HIV testing.

HIV testing was significantly higher among participants who belonged to the middle, richer, and richest wealth quintiles than those who belonged to the poorest among males. However, odds of HIV testing were lowers among those who were from the poorer, middle and richer as compared to the poorest in females. It meant that economic status was positively associated among males where as it was inversely associated among females. In contrast to the study, previous study repots respondents belonging to wealthier households were more likely to have been tested than those belonging to poorer households; additionally, wealth has a stronger effect among women than among men [7].

Male respondents who spent more than 1 month away from home in the last 12 months were more likely to be tested for HIV. However, this association was not significant among females. Similarly, labor migrants and their wives were more likely to have been tested for HIV than their nonmigrant counterparts in Nepal [9]. Respondents with multiple sexual partners were more likely to report HIV testing in both males and females. Given that having multiple sexual partners is the most important risk factor of contracting HIV, an increased utilization of HIV testing among them is one of the positive findings. The general population and the labor migrants are equally engaged in unprotected extramarital sex [9]. Additionally, delayed age at first sex might be linked with increased awareness on safe sexual practice, thereby further adhering to HIV testing, especially among females.

Mean HIV knowledge was 7.79 and 7.56 among males and females significantly. There was significant different in HIV knowledge between sexes. In the multivariate analysis, having comprehensive HIV knowledge was associated with increased likelihood of HIV testing among females only. This might signify that imparting comprehensive HIV knowledge work more for HIV testing in females than in males. Similar to the study, comprehensive HIV knowledge was significantly associated HIV testing in the females but not in the males in Zimbabwe [8]. Similarly, higher HIV knowledge was significantly associated with higher HIV testing in Burkina Faso according to the DHS survey [27]. A study conducted in Nepal using the NDHS 2011 also showed that social determinants were associated with poor HIV Knowledge among Nepalese males [28]. Thus, along with investing on education and economic status of the general population, comprehensive HIV knowledge should be promoted. The government of Nepal should consider the sociodemographic factors, mobility, sexual behavior and comprehensive HIV knowledge, while implementing public health intervention for HIV Prevention program. Furthermore, the HIV testing service should be integrated into the public health system of a government from center to community level in both urban and rural area.

### Limitations

The survey type may induce behavioral desirability bias. Individuals may be reticent or embarrassed to express their real sexual behaviors in face-to-face interviews. Hence, validating the respondents’ answers is challenging. The cross-sectional nature of the survey only allows for an association between variables of interest at the same time point. Thus, any cause-and-effect relationships could not be established. In addition, one of the limitations of the study is that we could use only the variables that were available in the data of the 2011 NDHS as factors associated with HIV testing in the model. Therefore, effect of some factors that might be associated with HIV testing such as blood donation and inappropriate needle use could not be assessed in the article.

## 5. Conclusions

The lifetime prevalence of HIV testing was relatively low, especially among females. In the adjusted model, age, ethnicity, education, occupation, economic status and HIV knowledge were significantly associated with lifetime HIV testing. Moreover, respondents with multiple sex partners and respondents who were away from home for more than 1 month in last 12 months were more likely to have been tested for HIV. Concern authority should consider imparting comprehensive HIV knowledge, socio-demographic variation and availability of the services at the grassroots level to promote HIV testing program.

## Figures and Tables

**Table 1 ijerph-16-03311-t001:** Measurement of Human immunodeficiency virus (HIV) knowledge based on ten questions.

S.N	Questions Asked	Coding
1	Have you heard about any infections that can be transmitted through sexual contact (STIs)?	Yes = 1, No = 0
2	Have you ever heard of an illness called AIDS?	Yes = 1, No = 0
3	Can people reduce their chance of acquiring the AIDS virus by having just one uninfected sex partner who has no other sex partners?	Yes = 1, No/Do not know = 0
4	Can people reduce their chance of acquiring the AIDS virus by using a condom every time they have sex?	Yes = 1, No/Do not know = 0
5	Can people acquire the AIDS virus from mosquito bites?	No = 1, Yes/Do not know = 0
6	Is it possible for a healthy-looking person to have the AIDS virus?	Yes = 1, No/Do not know = 0
7	Can one acquire HIV by sharing food with a person who has AIDS?	No = 1, Yes/Do not know = 0
8	Can HIV be transmitted from a mother to her baby during delivery?	Yes = 1, No/Do not know = 0
9	Can HIV be transmitted from a mother to her baby by breastfeeding?	Yes = 1, No/Do not know = 0
10	Are there any special drugs that a doctor or nurse can give to a woman infected with the AIDS virus to reduce the risk of transmission to the baby?	Yes = 1, No/Do not know = 0

**Table 2 ijerph-16-03311-t002:** Sociodemographic Characteristics of the Study Population (*N* = 12,860).

Variables	Categories	Male (*n* = 3017)		**Female (*n* = 9843)**	
		Number	Percentage/Mean (±SD)	**Number**	**Percentage/Mean (±SD)**
Age group (in years)	15–24	681	22.6	2470	25.1
	25–29	512	17.0	1982	20.1
	30–34	478	15.8	1664	16.9
	35–39	524	17.4	1544	15.7
	40–49	822	27.2	2183	22.2
Type of residence	Rural	2083	69.0	7138	72.5
	Urban	934	31.0	2705	27.5
Ethnicity	Others	1430	47.4	4416	44.9
	Upper caste	1200	39.8	4002	40.7
	Lower caste	387	12.8	1425	14.5
Religion	Hindu	2551	84.6	8443	85.8
	Buddha	266	8.8	798	8.1
	Others	200	6.6	602	6.1
Marital status	Others	393	13.0	397	4.0
	Married	2624	87.0	9446	96.0
Education	No schooling	473	15.7	4694	47.7
	Primary	720	23.9	1839	18.7
	Secondary	1336	44.3	2694	27.4
	Higher	488	16.2	616	6.3
Occupation	Unemployed	120	4.0	2047	20.8
	Skilled/unskilled manual	815	27.0	553	5.6
	Agriculture	963	31.9	5840	59.3
	Service *	1119	37.1	1403	14.3
Wealth quintile	Poorest	557	18.5	2008	20.4
	Poorer	511	16.9	1823	18.5
	Middle	562	18.6	1837	18.7
	Richer	599	19.9	1941	19.7
	Richest	788	26.1	2234	22.7
Mobility	No	1587	52.6	5948	60.4
	Yes	681	22.6	1055	10.7
	Missing	749	24.8	2840	28.9
Multiple sexual partners	Yes	1187	39.3	440	4.5
Age at first sex (in years)	-	2966	19.53 (±3.28)	9640	17.40 (±2.96)
HIV knowledge score	-	2760	7.79 (±1.22)	8249	7.56 (±1.32)
HIV testing	No	2474	82.0	9110	92.6
	Yes	543	18.0	733	7.4

* Professional/technical/managerial, HIV knowledge and Age at first sex were used as continuous variables.

**Table 3 ijerph-16-03311-t003:** HIV Testing by Sociodemographic and Behavioral Characteristics.

Variables	Categories	Males			Females		
		*n*	%/Mean (±SD)	*p* Value	*n*	%/Mean (±SD)	*p* Value
Age group (in years)	15–24	135	19.8	<0.001	220	8.9	<0.001
	25–29	117	22.9		209	10.5	
	30–34	106	22.2		125	7.5	
	35–39	93	17.7		97	6.3	
	40–49	92	11.2		82	3.8	
Type of residence	Rural	350	16.8	0.011	474	6.6	<0.001
	Urban	193	20.7		259	9.6	
Ethnicity	Others	218	15.2	0.000	189	4.3	<0.001
	Upper caste	254	21.2		385	9.6	
	Lower caste	71	18.3		159	11.2	
Religion	Hindu	465	18.2	0.305	673	8.0	<0.001
	Buddha	50	18.8		28	3.5	
	Others	28	14.0		32	5.3	
Marital status	Others	75	19.1	0.584	38	9.6	0.100
	Married	468	17.8		695	7.4	
Education	No schooling	26	5.5	<0.001	228	4.9	<0.001
	Primary	81	11.2		122	6.6	
	Secondary	307	23.0		256	9.5	
	Higher	129	26.4		127	20.6	
Occupation	Unemployed	29	24.2	<0.001	155	7.6	<0.001
	Manual	126	15.5		42	7.6	
	Agriculture	129	13.4		353	6.0	
	Service *	259	23.1		183	13.0	
Wealth quintile	Poorest	53	9.5	<0.001	150	7.5	<0.001
	Poorer	67	13.1		88	4.8	
	Middle	103	18.3		88	4.8	
	Richer	120	20.0		165	8.5	
	Richest	200	25.4		242	10.8	
Mobility	No	271	17.1	<0.001	436	7.3	<0.001
	Yes	168	24.7		112	10.6	
Multiple sexual partners	No	290	15.8	<0.001	689	7.3	0.037
	Yes	253	21.3		44	10.0	
HIV knowledge		530	8.06 (±1.10)	<0.001	716	8.16(1.16)	<0.001

*n* = number, % = percentage, * Professional/technical/managerial. HIV knowledge was used as continuous variable.

**Table 4 ijerph-16-03311-t004:** Multivariate Logistic Regression Analysis of Factors Associated with HIV Testing.

Variables	Categories	Males	*p* Value	Females	*p* Value
		AOR		AOR	
Age group (in years)	15–24	1		1	
	25–29	1.46 (1.03–2.06)	0.030	1.13(0.89–1.45)	0.304
	30–34	1.36 (0.95–1.96)	0.091	0.74 (0.55–1.00)	0.054
	35–39	1.14 (0.79–1.65)	0.465	0.53(0.37–0.76)	0.001
	40–49	0.78 (0.53–1.13)	0.190	0.47 (0.33–0.68)	<0.001
Type of residence	Rural	1		1	
	Urban	0.87 (0.66–1.16)	0.353	1.09 (0.87–1.37)	0.415
Ethicality	Other caste	1		1	
	Upper caste	1.20 (0.94–1.55)	0.136	1.94 (1.51–2.48)	<0.001
	Lower caste	1.53 (1.05–2.23)	0.027	3.57(2.63–4.83)	<0.001
Religion	Hindu	-	-	1	
	Buddha	-	-	0.84 (.52–1.36)	0.497
	Others	-	-	0.84 (.51–1.37)	0.495
Education	No schooling	1			
	Primary	1.50 (0.86–2.62)	0.153	0.86 (.63–1.170	0.344
	Secondary	2.82 (1.67–4.78)	<0.001	1.01(.75–1.37)	0.904
	Higher	2.85 (1.58–5.15)	<0.001	1.59 (1.06–2.37)	0.023
Occupation	Unemployed	1		1	
	Manual	1.23 (0.68–2.22)	0.481	1.09 (0.69–1.72)	0.697
	Agriculture	1.12 (0.62–2.04)	0.689	0.99 (0.74–1.31)	0.957
	Service *	1.28 (0.74–2.20)	0.371	1.58 (1.20–2.07)	0.001
Wealth quintile	Poorest	1		1	
	Poorer	1.33 (0.84–2.11)	0.215	0.51(0.35–0.73)	<0.001
	Middle	1.99 (1.28–3.07)	0.002	0.43(0.30–0.62)	<0.001
	Richer	1.68 (1.06–2.66)	0.025	0.67(0.47–0.95)	0.025
	Richest	2.35 (1.44–3.83)	0.001	0.73(0.49–1.07)	0.115
Mobility	No	1		1	
	Yes	1.68 (1.32–2.14)	<0.001	1.20 (0.94–1.54)	0.127
Multiple sexual partners	No	1		1	
	Yes	1.33 (1.05–1.68)	0.018	2.34 (1.54–3.56)	<0.001
Age at first sex		1.00 (0.96–1.04)	0.968	1.04 (1.01–1.08)	0.012
HIV knowledge		1.07 (0.96–1.18)	0.189	1.40 (1.29–1.52)	<0.001
Hosmer and Lemeshow Test (*p* value)		0.982		0.150	

1 indicates reference value, CIs: confidence intervals, AORs: Adjusted Odds Ratios, * Professional/technical/managerial, Age at first sex and HIV knowledge were used as continuous variables.

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
