# Peer review of "Role of Knowledge, Sociodemographic, and Behavioral Factors on Lifetime HIV Testing among Adult Population in Nepal: Evidence from a Cross-Sectional National Survey"

_ijerph, 2019, doi:10.3390/ijerph16183311_

Round 1

Reviewer 1 Report

General Comments:

This is an interesting study on a relevant topic for Nepal, where there are not many publications on HIV testing; the study is based on secondary data (DHS) and the authors used the adequate methods (logistic regression with complex sampling) to answer to the main objective of the study. However, to be published several issues must be considered and the English must be revised.

Specific Comments:

1.    Title: must indicate the type of study and/or the data used (DHS)

2.    Abstract: should have a fist paragraph on the context/ relevance of the study

3.    Introduction:

In the introduction the authors must explain the HIV situation in Nepal; is it a low or high risk country? What is the HIV prevalence and its main characteristics?

What is Nepal’s health profile? Most researchers know very little about Nepal, and this kind of information is crucial to understand the study’s context and to judge its potential contribution to current knowledge.

4.    Methods

In general, methods are well explained and without mistakes. However, the sentence in line 76 “Chi-square tests and t-test were utilized to assess the association between the explanatory variables and HIV testing” is not correct. T tests are not used to test associations. And later on, in the results section, we can´t find this t-test (only qui-square). The authors must explain or correct this sentence.

Another comment is that, in general, models are estimated separately for men and women, because literature suggests that men and women behave very differently in HIV issues. Usually papers on this subject (when data are from big data samples as it is for DHS), estimate two different models for men and women; In my opinion it will be very useful to have also these type of results.

5.    Results

In Table 2 I suggest to exclude the absolute number and to disaggregate data by sex.  It is always interesting to look at possible differences in men and women

In table 3, gender and age are repeated; the text starting at line 111 is related to Table 4 but it is below table 3. I think that it will be better if there is a paragraph on Table 3 results, then Table 4 and then the paragraph on

Table 4 results (otherwise we cannot understand why is Table 3 there).

In line 113- 114, the sentence “The highest odds ratio (OR) of HIV testing was observed among participants aged 24–29 years (AOR,114 2.17; CI, 1.61–2.90) and not those who aged 40–49 years” is not comprehensible.

Replaced it for example with: “The highest odds ratio (OR) for the odds of having been tested for HIV was observed among participants aged 24–29 years (AOR,114 2.17; CI, 1.61–2.90) when compared with aged 40–49 years”

Line 118, the sentence “Furthermore, HIV testing was significantly lower among those who had no formal schooling (AOR, 032; CI, 0.24–0.51)…. ……..“ is not correct. It should be: “Furthermore, the odds of being tested for HIV was significantly lower among those who had no formal schooling (AOR, 032; CI, 0.24–0.51)….

Line 123, same comment as before

Line 127, same comment as before

Line 129, the R2 statistic is this context is not very useful since the dependent variable is binary; use for example Hosmer and Lemeshow Test (it is available in SPSS)

6.    Discussion

Line 136. Include also the 95%CI related to the main outcome.

In my opinion, the results by sex must be estimate and then discussed here.

The English should be revised; many sentences difficult to understand.

7.    Conclusions

This section must be rewritten; the authors simply repeat the results. They should go further with specific public health recommendations.

Author Response

General Comments:

 This is an interesting study on a relevant topic for Nepal, where there are not many publications on HIV testing; the study is based on secondary data (DHS) and the authors used the adequate methods (logistic regression with complex sampling) to answer to the main objective of the study. However, to be published several issues must be considered and the English must be revised.

Answer:  Thank you so much for your time and effort to revise our manuscripts. This has surely improved the quality of our manuscript. The modifications we did are marked with red color in the manuscript.

Specific Comments:

1.    Title: must indicate the type of study and/or the data used (DHS)

 Answer: Thank you so much for such a useful advice. We have modified our title (title, page 1)

2.    Abstract: should have a fist paragraph on the context/ relevance of the study

Answer:  We have added one sentence about the relevance of the study in the abstract (Abstract. page: 1;lines: 12-13)

Introduction:

 In the introduction the authors must explain the HIV situation in Nepal; is it a low or high risk country? What is the HIV prevalence and its main characteristics?

Answer: We have added about the prevalence and its characteristics of HIV in the introduction section (Introduction, page: 1; lines:38-46)

What is Nepal’s health profile? Most researchers know very little about Nepal, and this kind of information is crucial to understand the study’s context and to judge its potential contribution to current knowledge.

 Answer: We have added main health profile of Nepal (Introduction, page 1-2;lines:46-50 )

4.    Methods

In general, methods are well explained and without mistakes. However, the sentence in line 76 “Chi-square tests and t-test were utilized to assess the association between the explanatory variables and HIV testing” is not correct. T tests are not used to test associations. And later on, in the results section, we can´t find this t-test (only qui-square). The authors must explain or correct this sentence.

 Thank you so much for pointing out our mistake. We have removed the t-test from analysis methods (Data Analysis, page: 3;line:110)

Another comment is that, in general, models are estimated separately for men and women, because literature suggests that men and women behave very differently in HIV issues. Usually papers on this subject (when data are from big data samples as it is for DHS), estimate two different models for men and women; In my opinion it will be very useful to have also these type of results.

Thank you so much for your suggestion. Separate analysis was performed for males and females. We performed reanalysis of the data disaggregating all the result by sex and wrote the description, abstract and discussion accordingly. (Table 2, 3, 4),   (Result, page: 4-8).

Results

In Table 2, I suggest to exclude the absolute number and to disaggregate data by sex.  It is always interesting to look at possible differences in men and women

Answer:  Yes, we have disaggregated data by sex (Table 2, page;4-5).

In table 3, gender and age are repeated; the text starting at line 111 is related to Table 4 but it is below table 3. I think that it will be better if there is a paragraph on Table 3 results, then Table 4 and then the paragraph on

 Answer:

Duplication of age and gender has been removed (Table 3). Description of table 3 is presented before table 3 (Result, page:5; ,line 146-150).

Table 4 results (otherwise we cannot understand why Table 3 is there).

In line 113- 114, the sentence “The highest odds ratio (OR) of HIV testing was observed among participants aged 24–29 years (AOR,114 2.17; CI, 1.61–2.90) and not those who aged 40–49 years” is not comprehensible.

Replaced it for example with: “The highest odds ratio (OR) for the odds of having been tested for HIV was observed among participants aged 24–29 years (AOR,114 2.17; CI, 1.61–2.90) when compared with aged 40–49 years”

Answer: This sentence is no more there. As separate models are made for males and females, discussion has been rewritten (Results, page:6 to 7; line 154-181).

Line 118, the sentence “Furthermore, HIV testing was significantly lower among those who had no formal schooling (AOR, 032; CI, 0.24–0.51)…. ……..“ is not correct. It should be: “Furthermore, the odds of being tested for HIV was significantly lower among those who had no formal schooling (AOR, 032; CI, 0.24–0.51)….

Answer: Thank you so much. We have rewritten this part as per new analysis (Results, page: 6 to 7; line 154-181).

Line 123, same comment as before

 Line 127, same comment as before

Answer: Thank you so much. We have rewritten this part as per new analysis. (Results, page: 6 to 7; line 154-181).

 Line 129, the R2 statistic is this context is not very useful since the dependent variable is binary; use for example Hosmer and Lemeshow Test (it is available in SPSS).

Answer: We have included the value of Hosmer and Lemeshow Test in place of R2 for both modes (Table 4).

Discussion

Line 136. Include also the 95% CI related to the main outcome.

In my opinion, the results by sex must be estimate and then discussed here.

Answer: now, we have now calculated results by sex and discussed accordingly (Discussion, page 8-9; lines: 190-272)

The English should be revised; many sentences difficult to understand.

Answer: We have modified some mistakes.

7.    Conclusions

 This section must be rewritten; the authors simply repeat the results. They should go further with specific public health recommendations.

Answer: We have modified conclusion (page: 10;  lines:282-288).

Reviewer 2 Report

Introduction section.

Please describe the full name in the first time (HIV, FSW, IDU, MSMs, PMTCT) and illustrate the mean of 90-90-90 targets. The article should show the current status or backgrounds of HIV in Nepal? Including prevalence, incidence, and mortality in the Introduction section. Please describe the motivation and significance of the article in the Introduction section. The article should add references regarding the influence factors of HIV screening in other countries. Do you have any theoretical framework to predict HIV screening?

Materials and methods

Please comment on the representative and data quality of NDHS Does the questionnaire have been verified by reliability and validity before the survey was conducted? There are not necessary to show all items of HIV screening in Table 1. You should mention the representative items of HIV, which can be accurately measured HIV knowledge. Items of HIV knowledge were measured by quantitative scale (such as Likert’s scale), not just use the binary variable. Generally, answers with no or do not know were not equal, it should separate in each item. The explanatory variables should define clearly in section 2.3, such as ecological region, developmental regions, wealth quintile, mobility etc. Multiple sexual partners were duplicated in Table 2. Schooling is replaced by education levels. HIV knowledge score should be measured by scores (means±SD), not shown by percentage Age at first sex (in years) is needed to explain, which means the age at first sexual experiences.

Results

Table 4 was analyzed by multiple or univariate logistic regression? Do you have test the association between explanatory variables? HIV knowledge was used by continuous variable or ordinal scale? HIV knowledge was positively associated with HIV testing, meaning higher HIV knowledge had a high percentage of HIV testing. Marital status had significance in Table 4, but OR=0.98 for the married group was similar to 1 for another group.

Discussion

The article should use the theory to explain the behavior of HIV testing and highlight the knowledge of HIV. We need to improve the knowledge or attitude and eventually modify the behavior of HIV testing, not focus on the demographic characteristics which may be related to HIV knowledge. If you have to measure the attitude or self-efficacy for all participants, the HIV testing will be fully explained by them. I recommended that you find the article highlight the link between HIV knowledge and behavior of HIV testing, also find explanatory variables to explain HIV knowledge. Based on the theory, there is a high correlation between knowledge, attitude, and behavior. Most explanatory variables were first predicted to knowledge but directly predict to the behavior of HIV testing. It is needed to add more references regarding risk factors for the knowledge and behavior of HIV. We are interested to the result of HIV testing and what are the risk factors for positive findings. Also, you can elaborate on each item of HIV knowledge. What item are a usual mistake? Why? How to improve HIV knowledge in the future. Limitations should highlight the theory is needed to explain your findings.      

Author Response

Answer:  Thank you so much for your time and effort to revise our manuscripts. This has surely improved the quality of our manuscript. The modifications we did are marked with red color in the manuscript.

Introduction section.

Please describe the full name in the first time (HIV, FSW, IDU, MSMs, PMTCT) and illustrate the mean of 90-90-90 targets. The article should show the current status or backgrounds of HIV in Nepal? Including prevalence, incidence, and mortality in the Introduction section. Please describe the motivation and significance of the article in the Introduction section. The article should add references regarding the influence factors of HIV screening in other countries. Do you have any theoretical framework to predict HIV screening?

Answer

Full form of HIV, FSW, IDU, MSMs, PMTCT are mentioned (Introduction, page-2,8; line:65-66,199) Meaning of 90-90-90 targets is included( page 2, line 58-61) Current status of HIV is included (Introduction, page-1-2; 38-46). For the theoretical model, we have included the following information in the manuscript “A theoretical framework considered for the study is, in addition to individual behavioral change and knowledge, the broad determinants of sexual behavior, such as gender, poverty, and mobility, should be focused in public health interventions” based on the reference of Wellings  et al, 2006 ( page: 2, line 62-64)

Materials and methods

Please comment on the representative and data quality of NDHS.  Does the questionnaire have been verified by reliability and validity before the survey was conducted? There are not necessary to show all items of HIV screening in Table 1. You should mention the representative items of HIV, which can be accurately measured HIV knowledge. Items of HIV knowledge were measured by quantitative scale (such as Likert’s scale), not just use the binary variable. Generally, answers with no or do not know were not equal, it should separate in each item. The explanatory variables should define clearly in section 2.3, such as ecological region, developmental regions, wealth quintile, mobility etc. Multiple sexual partners were duplicated in Table 2. Schooling is replaced by education levels. HIV knowledge score should be measured by scores (means±SD), not shown by percentage Age at first sex (in years) is needed to explain, which means the age at first sexual experiences.

Answer

NDHS 2011 is a nationally representative cross-sectional survey conducted under the government of Nepal. It is a part of worldwide Demographic and Health Survey (DHS) Program of USAID, which is carried out at every five years. this information is added in the manuscript (page 2, line76-78) The questionnaire was developed by DHS program and used internationally; for the survey, the questionnaire was translated into three major language of Nepal and was pretested. (page;2; lines: 82-84) Each item of HIV knowledge was measured as binary variables. Ten questions relevant to HIV knowledge were selected from the DHS questionnaire. Correct responses were coded as “1,” whereas incorrect or uncertain responses were coded as “0.” Items were summed to obtain the HIV knowledge score, with higher scores indicating more knowledge about HIV transmission and prevention. Thus, in logistic regression it summed value was used as continuous variables. It was performed based the reference of Yaya, et all, 2016. This is included in the manuscript. ( page: 3; lines:101-105) Schooling is replaced by education ( Table 2, table 3, table 4 and related description) Multiple sexual partners that were duplicated in Table 2 are deleted now. Measurement of some explanatory variables is explained. Two variables are excluded from the analysis (development region and ecological zone) (page:3; lines:93-100) HIV knowledge score and Age at first sex are presented as mean (±SD) ( table 2)

Results

Table 4 was analyzed by multiple or univariate logistic regression? Do you have test the association between explanatory variables? HIV knowledge was used by continuous variable or ordinal scale? HIV knowledge was positively associated with HIV testing, meaning higher HIV knowledge had a high percentage of HIV testing. Marital status had significance in Table 4, but OR=0.98 for the married group was similar to 1 for another group.

 Answer

Table 4 was analyzed by multiple logistic regressions. This is added in the table title. ( table 4) We did test the association between explanatory variables. Knowledge was measured by continuous scale (measurement of variables, page 3; line: 102-106) Yes, HIV knowledge was positively associated with HIV testing, meaning higher HIV knowledge had a high percentage of HIV testing (Measurement of HIV knowledge, page:3; line 104-105) As marital status was not significant in bivariate analysis, it was not now included in the multivariate analysis models for both male and female (Table 3 and table 4).

Discussion

The article should use the theory to explain the behavior of HIV testing and highlight the knowledge of HIV. We need to improve the knowledge or attitude and eventually modify the behavior of HIV testing, not focus on the demographic characteristics which may be related to HIV knowledge. If you have to measure the attitude or self-efficacy for all participants, the HIV testing will be fully explained by them. I recommended that you find the article highlight the link between HIV knowledge and behavior of HIV testing, also find explanatory variables to explain HIV knowledge. Based on the theory, there is a high correlation between knowledge, attitude, and behavior. Most explanatory variables were first predicted to knowledge but directly predict to the behavior of HIV testing. It is needed to add more references regarding risk factors for the knowledge and behavior of HIV. We are interested to the result of HIV testing and what are the risk factors for positive findings. Also, you can elaborate on each item of HIV knowledge. What item are a usual mistake? Why? How to improve HIV knowledge in the future. Limitations should highlight the theory is needed to explain your findings.      

 Answer:       

One of the limitations of the study is that we could use only the variables that were available in the data set provided by NDHS as factors associated with HIV testing in the model. This is added to the limitation now. (Limitation, page, 10, line: 278-280). We used previous article that shows HIV knowledge and socio-demographic factors is associated with HIV testing. (page:9; lines:260-268) Discussion part is rewritten as per new analysis for sex disaggregated analysis (discussion; page:8-9; lines: 190-272). New references are added to add the importance of HIV knowledge (discussion, page: 9; lines:260-68 Knowledge was used as continuous variable in descriptive statistics and in the multivariate analysis, thus proportion of individual item was not done (table 4).

Round 2

Reviewer 1 Report

The authors addressed all my issues.

The paper is now ready to be published

Author Response

Thank you so much for your time and effort to review our manuscript.  This has surely improved the quality of this manuscript.

Reviewer 2 Report

Generally, the article has improved and some questions have responded clearly. But some points should be described based on the theoretical frame.

1. We know the KAP (Knowledge, attitude, and behavior) was used to validate or predict the behavior. In the study, HIV knowledge should test the association with HIV testing was behavior. Also,  sociodemographic information have significantly associated with HIV knowledge. sociodemographic information directly linked to HIV knowledge and then associated with HIV testing. Therefore, Table 3 should show the association between HIV knowledge and socio-demographic information.  Table 4 just emphasize the different level of HIV knowledge associated the HIV testing behavior. Because of the HIV knowledge may be improved and further do healthy behaviors. Other problems in the article, there are intercorrelated between socio-demographic information. If you input in the model, it leads multicollinearity, such as education, occupation, wealth and type of residence, etc.    

2. Unfortunately,  HIV testing behavior is not only contributing to sexual partners citizens, other factors including blood donation, parental vertical heredity or inappropriate needle use, etc. The article should elaborate on the points.

3. The results in this article were stratified by gender. It is needed to discuss the gender-difference of HIV knowledge and testing in the article. 

Author Response

Thank you so much for your time and effort to review our manuscript.  This has surely improved the quality of this manuscript. The revision we did are marked with red color.

Comments and Suggestions for Authors

Generally, the article has improved and some questions have responded clearly. But some points should be described based on the theoretical frame.

We know the KAP (Knowledge, attitude, and behavior) was used to validate or predict the behavior. In the study, HIV knowledge should test the association with HIV testing was behavior. Also,  sociodemographic information have significantly associated with HIV knowledge. sociodemographic information directly linked to HIV knowledge and then associated with HIV testing. Therefore, Table 3 should show the association between HIV knowledge and socio-demographic information.  Table 4 just emphasize the different level of HIV knowledge associated the HIV testing behavior. Because of the HIV knowledge may be improved and further do healthy behaviors. Other problems in the article, there are intercorrelated between socio-demographic information. If you input in the model, it leads multicollinearity, such as education, occupation, wealth and type of residence, etc.   

Answers:

Association between HIV knowledge and HIV testing was tested by t test and added in the table 3 (table 3). We fully appreciate your concern; socio-demographic variables might be associated with HIV knowledge. However, the objective of the study was to assess the effects of HIV knowledge on HIV testing along with other socio-demographic variables. Thus, we did not calculate the association between socio-demographic factors and HIV knowledge. In spite of this, multicollinearity was checked; variance inflation factor (VIF) for each variable with HIV knowledge was less than 2. This information has been included in the data analysis section (Data analysis, page 4, line 115-116). Unfortunately, HIV testing behavior is not only contributing to sexual partners citizens, other factors including blood donation, parental vertical heredity or inappropriate needle use, etc. The article should elaborate on the points.

Answer: Thank you so much for your concern. As we used secondary data, we could not assess other factors including blood donation, vertical transmission and inappropriate needle use. Thus, we have now added this information in the limitation section (Limitation, page 10, 287-289)

The results in this article were stratified by gender. It is needed to discuss the gender-difference of HIV knowledge and testing in the article. 

Answer: Thank you so much for your suggestion.  We have added the gender-difference of HIV knowledge also in the article (Discussion, page 10, 264-268).
